# Pretraining on the Test Set Is No Longer All You Need: A Debate-Driven Approach to QA Benchmarks

**Linbo Cao** [*†]
Faculty of Mathematics
University of Waterloo
Waterloo, ON, Canada
l6cao@uwaterloo.ca

**Jinman Zhao** [*]
Department of Computer Science
University of Toronto
Toronto, ON, Canada
jzhao@cs.toronto.edu

## Abstract

As frontier language models increasingly saturate standard QA benchmarks, concerns about data contamination, memorization, and escalating dataset creation costs persist. We propose a debate-driven evaluation paradigm that transforms any existing QA dataset into structured adversarial debates—where one model is given the official answer to defend, and another constructs and defends an alternative answer—adjudicated by a judge model blind to the correct solution. By forcing multi-round argumentation, this approach substantially increases difficulty while penalizing shallow memorization, yet reuses QA items to reduce curation overhead. We make two main contributions: (1) an evaluation pipeline to systematically convert QA tasks into debate-based assessments, and (2) a public benchmark that demonstrates our paradigm's effectiveness on a subset of MMLU-Pro questions, complete with standardized protocols and reference models. Empirical results validate the robustness of the method and its effectiveness against data contamination—a Llama 3.1 model fine-tuned on test questions showed dramatic accuracy improvements (50% → 82%) but performed worse in debates. Results also show that even weaker judges can reliably differentiate stronger debaters, highlighting how debate-based evaluation can scale to future, more capable systems while maintaining a fraction of the cost of creating new benchmarks. Overall, our framework underscores that "pretraining on the test set is no longer all you need," offering a sustainable path for measuring the genuine reasoning ability of advanced language models.

## 1 Introduction

**Benchmark saturation** is a critical challenge in NLP evaluation (Ott et al., 2022), driven by rapid advances in large language models (LLMs) such as GPT-4 (OpenAI, 2023; Bubeck et al., 2023), OpenAI's o1 (OpenAI, 2024c), and DeepSeek-R1 (DeepSeek-AI, 2025). Continuous model improvements quickly exhaust existing benchmarks, evidenced by the progression from GLUE (Wang et al., 2019b) to SuperGLUE (Wang et al., 2019a), and from MMLU (Hendrycks et al., 2021) to MMLU-Pro (Wang et al., 2024), as well as the emergence of tougher tests like ARC-AGI (Chollet, 2019; Chollet et al., 2024) and HLE (Phan et al., 2025). Such developments highlight fundamental evaluation challenges as AI capabilities approach artificial general intelligence (AGI) or artificial superintelligence (ASI). Another major issue, **data contamination**, occurs when models achieve artificially inflated scores due to training on benchmark test data, as illustrated by Schaeffer (2023). Comprehensive reviews by Sainz et al. (2023) emphasize this issue, prompting solutions like LiveBench (White et al., 2025), which regularly introduces new questions to minimize contamination, and interactive evaluations like KIEval (Yu et al., 2024) that distinguish memorized responses from true understanding. Resource limitations further complicate benchmark creation, exemplified by

---

[*]Equal contribution.
[†]Corresponding author.

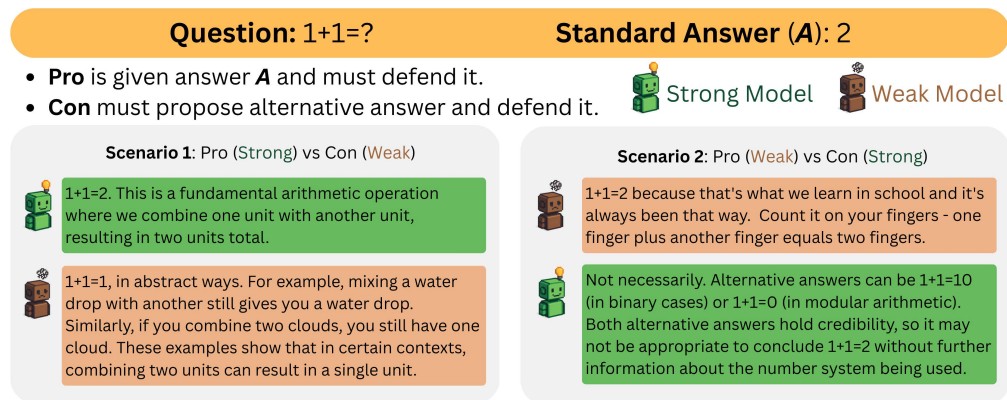

Figure 1: Illustration of how debates might encourage deeper reasoning, distinguishing strong and weak models beyond superficial correctness.

HLE's reliance on nearly 1000 experts from over 500 institutions (Phan et al., 2025), making such efforts increasingly unsustainable.

Current evaluation alternatives include holistic frameworks such as HELM (Liang et al., 2023), offering multidimensional metric analyses, and head-to-head evaluations like Chatbot Arena (Chiang et al., 2024), though it might suffer from inconsistent evaluation criteria. Automated assessments with LLM-based judges, such as Auto-Arena (Zhao et al., 2024), face reliability challenges. Existing debate-based evaluations, including FlagEval (Beijing Academy of Artificial Intelligence, 2025) and methods from Bandi & Harrasse (2024), often lack standardization, typically relying on open-ended questions (e.g., MT-Bench (Zheng et al., 2023)). Our method addresses this gap by converting structured QA tasks into adversarial debates, utilizing official answers to facilitate objective assessment, reducing biases common in open-ended evaluations.

Our core approach, **debate-driven QA evaluation**, modifies existing QA tasks by retaining only questions and correct answers, removing incorrect alternatives. A *Pro* model supports the official answer, while a *Con* model proposes and defends an alternative. Keeping all original options would compel defending obviously incorrect choices, creating alignment issues; conversely, fully open-ended formats risk amplifying evaluator bias if both debaters produce incorrect solutions. This structured debate system integrates smoothly with existing datasets, featuring double round-robin formats with role reversals to eliminate positional bias. Assigning explicit **adversarial roles** incentivizes deeper reasoning, penalizing superficial memorization and requiring coherent logical arguments, as illustrated in Figure 1. Finally, an **answer-blind judge** assesses arguments purely based on quality, prioritizing genuine reasoning over memorized answers.

We introduce a systematic, reproducible evaluation pipeline transforming standard QA datasets into structured adversarial debates, effectively reducing subjective biases, clearly signaling model reasoning capabilities, and significantly decreasing benchmark curation costs. Empirical experiments with various commercial and open-source models on MMLU-Pro (Wang et al., 2024) validate our method's robustness and resistance to data contamination. Additionally, weaker judges reliably distinguished stronger debaters, reinforcing findings by Khan et al. (2024), thereby ensuring scalability and future-proof applicability for advanced AI evaluations. We plan to publicly release a standardized benchmark derived from 5,500 structured debates—totaling over 11,000 rounds of argumentation—involving eleven reference models, providing the research community with a dependable evaluation resource. [1]

---

[1]Code for this project is available at: https://github.com/l6cao/Debate-Driven-Evaluation.

## 2 Related Work

**Traditional QA Benchmarks and Limitations.** The evolution of QA benchmarks has witnessed increasing saturation as language models rapidly advance. Early benchmarks like GLUE (Wang et al., 2019b) quickly led to SuperGLUE (Wang et al., 2019a), while MMLU (Hendrycks et al., 2021) spawned more challenging variants including MMLU-Pro (Wang et al., 2024), MMLU-Pro+ (Taghanaki et al., 2024), and C-MMLU (Li et al., 2024b). Other specialized benchmarks emerged, such as GSM8K (Cobbe et al., 2021) for mathematical reasoning, GPQA (Rein et al., 2023) for expert-level questions, and BIG-Bench (Srivastava et al., 2023) with over 200 diverse tasks. Advanced models including OpenAI's GPT-4 (OpenAI, 2023), Claude 3.5 Sonnet (Anthropic, 2024), Google's Gemini (Pichai et al., 2024), and open-source alternatives like DeepSeek-V3 (DeepSeek-AI, 2024), and recent advancements in reasoning models such as OpenAI's o1 (OpenAI, 2024c) and DeepSeek-R1 (DeepSeek-AI, 2025), employ chain of thought (Wei et al., 2022), demonstrating performance gain with sound reasoning (Bi et al., 2025). Concurrently, the community's enthusiasm for enhancing LLM reasoning has remained high, yielding significant new methods continuously, such as Chain-of-Thought prompting (Wei et al., 2022), Self-Consistency (Wang et al., 2023), Multi-Agent Debate (Liang et al., 2024), and deductive-inductive frameworks (Cai et al., 2025). Such models and methods post threat to saturation of these benchmarks, necessitating increasingly difficult challenges such as ARC-AGI (Chollet, 2019), ARC-AGI-2 (ARC Prize Foundation & Kamradt, 2025), Humanity's Last Exam (Phan et al., 2025), and FrontierMath (Glazer et al., 2024). Economic constraints further compound these challenges—BIG-Bench involved 450 authors across 132 institutions, HLE mobilized nearly 1000 subject experts from over 500 institutions, and FrontierMath assembled 60+ mathematicians including IMO gold medalists—highlighting the unsustainability of frequent benchmark creation.

Data contamination represents another critical challenge to benchmark integrity. Sainz et al. (2023) highlight how test set leakage threatens evaluation validity, while Schaeffer (2023) satirically demonstrate the possibility of achieving perfect scores by directly training on test data. Studies by Balloccu et al. (2024), Golchin & Surdeanu (2024), and Xu et al. (2024) document widespread contamination issues, especially in closed-source models, undermining the reliability of performance comparisons. To this end, researchers have developed sophisticated detection techniques, such as showing that models can guess masked test answers (Deng et al., 2024) or using statistical tests to prove contamination from model outputs (Oren et al., 2024; Shi et al., 2024). Researchers have proposed various solutions to address this problem, including continuous benchmark refreshes via LiveBench (White et al., 2025), filtering contaminated examples (Gupta et al., 2024), and dynamic evaluation protocols like KIEval (Yu et al., 2024) and DyVal (Zhu et al., 2024). However, these approaches either rely on expensive private datasets or cannot fully resolve contamination in established benchmarks, leaving significant challenges that our debate-driven methodology directly addresses.

**Multi-Agent Debate (MAD).** Multi-agent debate originated from *AI Safety via Debate* (Irving et al., 2018), introducing adversarial dialogues where agents advocate positions evaluated by human judges. Recent studies like Liang et al. (2024) extended this to foster divergent reasoning in LLMs. Debates help achieve *truth emergence*, with Khan et al. (2024) showing debates among advanced models enable weaker judges to better discern truthful answers, while Lang et al. (2025) demonstrated debate's capacity to enhance weak-to-strong model alignment and reduce hallucinations. Du et al. (2024) and Li et al. (2024c) confirmed iterative model interactions significantly mitigate LLM hallucinations. The success of debate frameworks depends on *judge models*—Zheng et al. (2023) found LLM-based judges effectively approximate human judgment, Liu et al. (2024) confirmed LLM judges can surpass human accuracy but warned of lexical biases, and Khan et al. (2024) showed even weaker judges can effectively evaluate stronger debaters. Implementation variants include Bandi & Harrasse (2024)'s courtroom-style multi-agent debate, Moniri et al. (2024)'s automated model ranking, Beijing Academy of Artificial Intelligence (2025)'s broader framework, and Rahnamoun & Shamsfard (2025)'s multi-layered metrics. Unlike existing approaches, our work specifically adapts debates to *structured QA tasks*, addressing data contamination while ensuring clear reasoning assessments.

**Dynamic Evaluation Methods.** Dynamic evaluation methods offer flexible assessments of language models beyond static benchmarks. Chatbot Arena (Chiang et al., 2024) exemplifies head-to-head comparison through user-driven evaluation, though with inconsistent judging criteria. LLM-based automated evaluation has emerged as an alternative—Auto-Arena (Zhao et al., 2024) utilizes LLM judges for model comparison, FlagEval (Beijing Academy of Artificial Intelligence, 2025) implements debate frameworks, KIEval (Yu et al., 2024) focuses on distinguishing understanding from memorization, and others use structured or Socratic questioning to evaluate discourse comprehension and faithfulness (Wu et al., 2023; Miao et al., 2024). Language-Model-as-an-Examiner (Bai et al., 2023) also creates adaptive questions. Comprehensive surveys by Li et al. (2024a) and Gu et al. (2024) examine such LLM-as-judge systems. These approaches require robust ranking mechanisms like Elo (Elo, 1978) for skill progression, Bradley–Terry models (Bradley & Terry, 1952) for probabilistic comparison, and TrueSkill (Herbrich et al., 2006) for Bayesian inference. Despite their advantages, these methods struggle with reproducibility—variations in prompts, match ordering, and evaluator biases yield inconsistent results. Current research aims to balance flexibility with consistency for reliable model evaluation.

# 3 Debate-Based Evaluation Framework

This section describes our methodology for adapting QA benchmarks into structured debates, systematic evaluation protocols, and standardized benchmark creation from debate outcomes for efficient model comparisons.

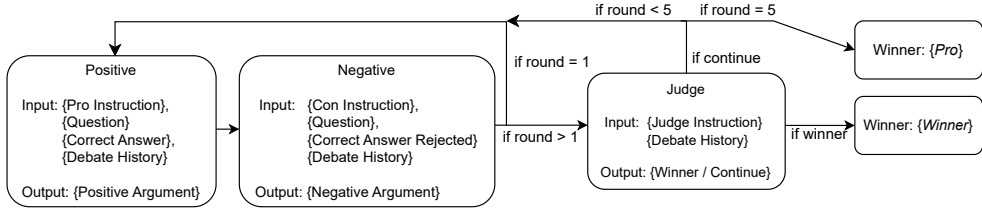

Figure 2: Single debate pipeline showing question transformation, role assignment, multi-round exchanges, and blind judging process.

**Debate Overview:** Our framework converts QA datasets with clearly defined answers, such as MMLU-Pro (Wang et al., 2024), into structured debates by removing multiple-choice distractors. This grants the Con side freedom to propose alternatives while preserving the original correct answer for the Pro side. Keeping all original distractors would force debaters into defending obviously incorrect answers, creating alignment concerns, while entirely open-ended formats could amplify judge bias when both sides are incorrect.

Our pipeline assigns distinct roles: the Pro model defends the provided correct answer, and the Con model, instructed that the official answer is incorrect, proposes and defends an alternative. This adversarial approach incentivizes deeper reasoning rather than memorization. Debates consist of multiple rounds (2-5), a range chosen to balance argumentative depth with computational efficiency. This choice is informed by prior work (Liang et al., 2024; Du et al., 2024), which found diminishing returns in debate quality beyond this scope. A minimum of two rounds mitigates judge biases, particularly in data contamination scenarios. If no clear winner emerges after five rounds, the Pro side is awarded victory for demonstrating sustained defense. Figure 2 illustrates the debate pipeline, including question transformation, role assignment, multi-round exchanges, and blind judging.

Judges evaluate debates blindly, seeing only the question and debate transcript, and return verdicts as "Positive" (Pro wins), "Negative" (Con wins), or "Continue" (another round required), ensuring unbiased evaluation based on argument quality alone.

**Evaluation Protocols:** Our evaluation employs a double round-robin format, where each model debates all others in both Pro and Con roles to mitigate positional biases. Models

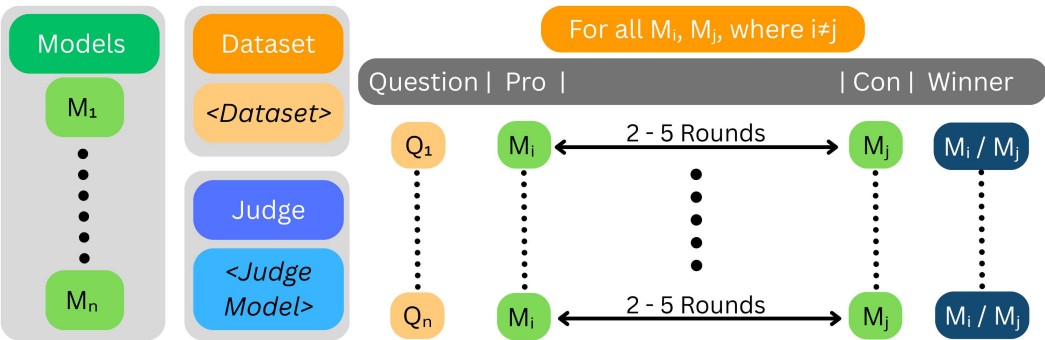

Figure 3: Double round-robin evaluation process showing model pairings, role alternation, and aggregate scoring system.

effective at defending correct answers may struggle in adversarial roles, and contamination may artificially enhance defensive performance. Figure 3 depicts the double round-robin process, including model pairings, role switching, and scoring methods.

The primary evaluation metric is total win count, enabling reproducible, order-independent rankings. Debate results are logged for analysis and future use.

**Benchmark Creation:** We create standardized benchmarks through reference debates between established models, reusable in subsequent evaluations. New models only debate against selected reference models in both roles, supplemented by stored reference debates. Introducing one new model at a time preserves integrity, enabling consistent scoring across evaluations.

For ranking, we examine various methods: simple win counts (complete tournaments required), Elo ratings (Elo, 1978) (progressively updated scores), Bradley-Terry models (Bradley & Terry, 1952) (probabilistic pairwise comparisons), and TrueSkill (Herbrich et al., 2006) (Bayesian rankings suited for incomplete comparisons). Our approach first prioritizes stability and usability of reference model scores, ensuring minimal fluctuation as new models are introduced, and subsequently emphasizes producing benchmark outcomes that are globally comparable.

This framework supports reproducible, consistent evaluation across implementations, allowing reliable comparison of model reasoning without creating additional QA datasets.

## 4 Experimental Setup

We randomly sampled 50 questions from the MMLU-Pro benchmark (Wang et al., 2024) as our evaluation dataset. Our main experiment involved eleven diverse models: *DeepSeek V3* (DeepSeek-AI, 2024), *Claude 3.5 Sonnet* (Anthropic, 2024), *GPT-4o* (OpenAI, 2024a), *GPT-4o mini* (OpenAI, 2024b), *GPT-3.5-turbo*[2], *Claude 3.5 Haiku* (Anthropic, 2025), *Mistral Large* (MistralAI, 2025), *Mixtral 8×7B* (Jiang et al., 2024), *Mixtral 8×22B* (MistralAI, 2024), *Mistral 7B* (Jiang et al., 2023), and *Llama 3.1 8B* (Grattafiori et al., 2024). For our main experiment, we used *GPT-4o* as the judge model. To confirm the generalizability of our method, we also conducted an evaluation on the GPQA main test set (Rein et al., 2023) (448 questions) using several open-source models, with the full setup detailed in Appendix C.

To assess data contamination effects, we performed a fine-tuning analysis of *Llama 3.1 8B* using LoRA (Hu et al., 2022) trained explicitly on the test set. We also conducted a separate experiment to test judge model variation, using seven different judges: *Mistral Large*, *GPT-4o*, *GPT-4o mini*, *Mixtral 8×7B*, *Mistral 7B*, along with the original *Llama 3.1 8B* and its newly fine-tuned variant. Each debate followed our structured protocol (Section 3)

---

[2] https://platform.openai.com/docs/models

Table 1: QA Accuracy (50 MMLU-Pro 0-shot CoT) vs. Debate Win Counts.

| Model | QA Acc (%) | QA Rank | Debate Wins | Debate Rank | Δ |
|---|---|---|---|---|---|
| Claude 3.5 Sonnet | 80 | 1 | 718 | 2 | -1 |
| DeepSeek V3 | 74 | 2 | 759 | 1 | +1 |
| GPT-4o | 72 | 3 | 581 | 5 | -2 |
| Mistral Large | 70 | 4 | 636 | 3 | +1 |
| GPT-4o mini | 62 | 5 | 564 | 6 | -1 |
| Claude 3.5 Haiku | 60 | 6 | 621 | 4 | +2 |
| Mixtral 8×22B | 56 | 7 | 385 | 8 | -1 |
| Llama 3.1 8B | 50 | 8 | 348 | 9 | -1 |
| GPT-3.5-turbo | 48 | 9 | 263 | 10 | -1 |
| Mixtral 8×7B | 38 | 10 | 395 | 7 | +3 |
| Mistral 7B | 34 | 11 | 230 | 11 | 0 |

with a double round-robin format, aggregating model win counts as the primary metric. For benchmark standardization, we included our dataset of 5,500 debate transcripts—which collectively contain over 11,000 argumentative rounds—spanning the full double round-robin interactions among the eleven reference models, tested various scoring methods including Elo, Bradley–Terry, and TrueSkill, and ultimately selected TrueSkill due to its robustness on partial data.

## 5 Results and Analysis

### 5.1 Overall Performance Comparison

**Traditional QA Accuracy.** We first measure standard QA accuracy for all eleven models on 50 MMLU-Pro (0-shot CoT) questions. Each model's final answer was checked for exact match with the official solution in a single-pass format. Despite notable gaps—with *Claude 3.5 Sonnet* scoring 80% and *Mistral 7B* trailing at 34%—several models still achieve relatively high scores, suggesting partial saturation even under conventional single-turn QA.

**Debate Win Counts.** We next conduct multi-round, double round-robin debates, with models alternating between defending (Pro) and questioning (Con) the official answers. Debate outcomes, summarized in Table 1, show generally strong correlation with single-turn QA accuracy—most models shift by only one or two positions. However, debates add valuable nuance: *Claude 3.5 Haiku* climbs two ranks due to superior argumentation, while *GPT-4o* drops slightly, highlighting that debate evaluation effectively captures deeper reasoning skills beyond standard QA performance.

We further analyze debate performance by isolating results from models exclusively in the defender (Pro) role, reflecting their ability to justify and maintain correct solutions. Figure 4 compares total debate wins (combined Pro and Con) to Defending-Only Wins and Questioning-Only Wins, illustrating the consistency of model rankings across these roles.

### 5.2 Role-Specific Performance Analysis

The rankings based solely on Defending-Only Wins perfectly align with overall debate win rankings for the top nine models, underscoring the robustness of our evaluation framework. The minor discrepancy observed between *GPT-3.5-turbo* and *Mistral 7B* at the bottom is negligible, indicating that a model's overall debate success consistently reflects its genuine understanding and problem-solving capabilities.

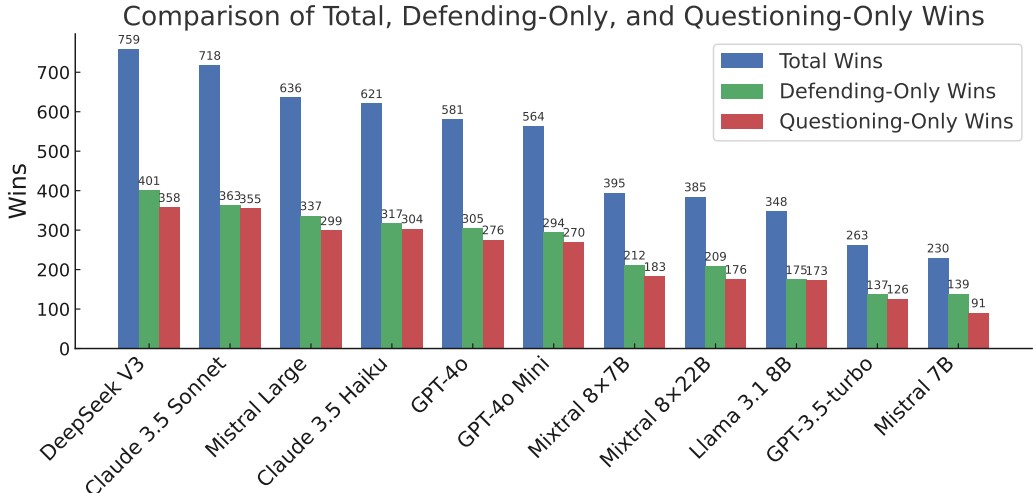

Figure 4: Comparison of Total Debate Wins vs. Defending-Only and Questioning-Only Wins.

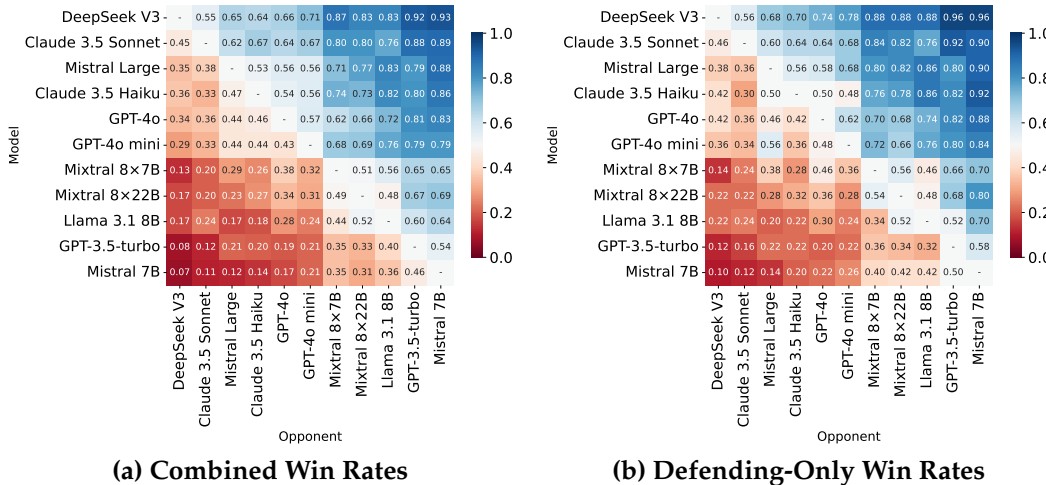

**(a) Combined Win Rates**

**(b) Defending-Only Win Rates**

Figure 5: Pairwise head-to-head win rate heatmaps across all 11 models. Each cell $(i, j)$ shows the win rate of model $i$ against model $j$. (a) combines Pro and Con roles, while (b) focuses only on defending correct answers. These reveal clear ranking structure and robustness against positional and contamination effects.

## 5.3 Head-to-Head Model Comparisons

**Pairwise Heatmaps and Transitivity Analysis.** We constructed pairwise win rate heatmaps across three scenarios—questioning, defending, and overall—to refine the rankings in Table 1. Stronger models consistently dominated weaker ones, typically securing 70–90% win rates. While a few minor loops were observed in the defending-only and questioning-only settings, notably, the combined heatmap resolved almost all of these inconsistencies. Out of 55 unique model pairings, the combined results produced a remarkably consistent hierarchy, with only a single, marginal transitivity violation. This near-perfect (98%+) transitivity strongly suggests our role-switching approach effectively mitigates positional bias and data contamination, where contaminated models may artificially excel in defense roles. This high degree of fidelity implies our evaluation genuinely captures meaningful differences in models' reasoning capabilities. For instance, even when provided with

the correct answer, the weaker model (Mistral 7B) achieved only a 10% win rate against DeepSeek V3, clearly underscoring our framework's effectiveness in distinguishing true model competence. Furthermore, this high level of consistency was confirmed in our supplementary evaluation on the GPQA benchmark (Appendix C), where the rankings exhibited perfect transitivity, underscoring the robustness of the method across different datasets.

**Partial Tournament Validity.** A key finding from our large-scale experiments is the near-perfect transitivity of debate outcomes. As evidenced by both the combined win rate heatmap (Figure 5a) and our confirmatory evaluation on the full GPQA dataset (Appendix C)—which seldom showed intransitive loops—a clear and stable performance hierarchy consistently emerges. This high degree of consistency has a critical practical implication: if a new model is evaluated and found to win against a reference model A but lose to model B, we can conclude with high confidence that its capability lies within the range of A to B. This property enables a highly efficient, binary search-like evaluation protocol. In practice, this shifts the computational requirement for benchmarking a new model from a costly linear scan against all reference models (O(n)) to a much more manageable logarithmic one (O(log n)), where n represents the number of reference models in the debate benchmark set.

### 5.4 Fine-Tuning Impact Assessment

Table 2: Debate performance (Win Rate) of Llama 3.1 8B before and after fine-tuning. *Theoretical win rate for a self-match; role-specific rates (–) are not applicable.

| Model | vs. Llama 3.1 8B | | | vs. DeepSeek V3 | | |
|---|---|---|---|---|---|---|
| | Overall | Defending | Questioning | Overall | Defending | Questioning |
| Llama 3.1 8B | 0.50* | – | – | 0.17 | 0.22 | 0.12 |
| Llama 3.1 8B Finetuned | 0.46 ↓ | 0.48 ↓ | 0.44 ↓ | 0.16 ↓ | 0.26 ↑ | 0.06 ↓ |

We conducted a fine-tuning experiment using LoRA on the Llama 3.1 8B model with the evaluation set, resulting in a significant accuracy improvement from 50% to 82%. However, this improvement in standard QA did not translate to enhanced debate performance (Table 2). Compared to its non-fine-tuned baseline, the fine-tuned model performed worse overall against both the original Llama 3.1 8B (0.50→0.46) and the state-of-the-art DeepSeek V3 (0.17→0.16), particularly suffering a substantial drop in questioning ability (0.12→0.06).

The disparity between improved static accuracy and decreased debate efficacy suggests that fine-tuning primarily enhanced memorization without developing deeper comprehension, aligning with similar findings by KIEval (Yu et al., 2024). Crucially, our debate-based approach exposes and mitigates such contamination effects, effectively distinguishing genuine understanding from superficial recall. This demonstrates that our method robustly evaluates model reasoning capabilities without requiring dataset filtering or the creation of new benchmarks.

### 5.5 Judge Model Variations

**Multi-Judge Experiment.** To assess judge model variation, we conducted seven independent debate tournaments. Each tournament featured the same four debater models (*Mistral Large*, *GPT-4o Mini*, *Mixtral 8×7B*, and *Mistral 7B*) but was adjudicated by one of seven distinct judges, including one contaminated model (*Llama 3.1 8B FT*). Table 3 summarizes the cumulative win counts from each of these judge-led tournaments (pairwise results provided in Appendix Figure 7 and Table 4). Despite varied judging capabilities, six of the seven judges—including the contaminated model—produced identical debater rankings: *Mistral Large > GPT-4o Mini > Mixtral 8×7B > Mistral 7B*, confirming the robustness of the debate outcomes across different adjudicators.

**Rank Consistency and Judge Limitations.** Six judges, including the contaminated one, as evidenced by the pairwise heatmaps (Figure 7), yield entirely consistent rankings with

Table 3: Total win counts for each debating model (*columns*) as evaluated by different judge models (*rows*). Judges are ordered by capability (strongest to weakest). The rankings of debaters remain identical, with the exception of the weakest judge (Mistral 7B), which failed to differentiate between models.

| Judge Model \ Debater | Mistral Large | GPT-4o Mini | Mixtral 8×7B | Mistral 7B |
|---|---|---|---|---|
| Mistral Large | 209 | 176 | 122 | 93 |
| GPT-4o | 215 | 191 | 126 | 68 |
| GPT-4o Mini | 233 | 192 | 114 | 61 |
| Mixtral 8×7B | 202 | 190 | 127 | 81 |
| Llama 3.1 8B | 167 | 162 | 144 | 127 |
| Llama 3.1 8B Finetuned | 169 | 167 | 147 | 117 |
| Mistral 7B | 150 | 151 | 150 | 149 |

minimal intransitive results (e.g., no significant cycles such as A>B>C>A). However, *Mistral 7B* exhibited poor discrimination because it appears to lack sufficient long-context evaluation and instruction-following capability. It consistently failed to output the required judgment formats, causing the system to keep falling back to the default positive option. Thus, we conclude that limited by its instruction-following capabilities, it is not yet capable of judging. Yet, given the closely aligned results from the remaining judges, we may conclude the ranking is fairly robust and consistent across judge models, as long as they possess sufficient instruction-following capabilities.

**Superintelligence Readiness.** Importantly, our experiments support conclusions from Khan et al. (2024), demonstrating weaker models can still reliably evaluate stronger ones within structured debate settings. Unlike conventional QA tasks with fixed performance ceilings, our framework's adversarial nature provides a theoretically unlimited measurement range. Consequently, even future superintelligent models would be meaningfully assessed, as success in debates necessitates robust and explicit reasoning that weaker judges can effectively evaluate.

## 5.6 Ranking Algorithm Comparison

**Why Not Simple Win Counts, Elo, or Bradley–Terry?** A naive approach is to rank by total wins, but this requires a complete double round robin and is unsuited for incremental participation. Elo simplifies sequential updates yet suffers from order dependence and partial-match instability. Bradley–Terry (BT) fits pairwise outcomes via logistic regression, which reduces sequence effects; however, as shown in Tables 6 and 7, even a single new model can shift reference-model scores drastically (e.g., DeepSeek V3 from 1963.0 to 1981.0), undermining benchmark stability.

**TrueSkill for Stability.** TrueSkill uses a Bayesian factor-graph framework to handle skill inference in tournaments with incomplete or disordered match data. Our chosen parameters $\mu = 25$, $\sigma = 8.333$, $\beta = 4.5$, $\tau = 0.01$ consistently align scores with observed win counts, while limiting rating shifts when new participants appear (e.g., DeepSeek V3's score moving only 492.0 to 489.4). Such minimal disturbance preserves previously established orders and ensures reliable integration of newly added models.

**Practical Considerations.** Real-world leagues rarely conduct full round robins; new models enter on rolling bases, some matchups remain unplayed, and computational resources vary. TrueSkill's partial-match capability ensures that incomplete data still yields coherent rankings, where established references are only marginally affected by new arrivals. Consequently, our benchmark employs TrueSkill as the primary ranking mechanism, guaranteeing consistent, reproducible standings across both partial and complete tournaments.

# 6 Discussion and Conclusion

Our experiments have demonstrated that debate-driven evaluation provides a robust alternative to traditional QA benchmarks, addressing key challenges in AI assessment. The framework's effectiveness is validated by several key findings: highly transitive rankings (98%+ consistency) across a diverse set of models, aligned results across different judge models, and reliable detection of shallow memorization in contaminated models.

Our approach directly addresses **data contamination**, showing that while fine-tuning can artificially inflate traditional QA performance (50% → 82%), it fails to improve—and can even harm—performance in debate scenarios. This aligns with findings from KIEval (Yu et al., 2024), confirming that debate formats successfully distinguish genuine understanding from memorization. Furthermore, our method offers a remedy for **benchmark saturation** without creating entirely new datasets—a crucial advantage as creating high-quality benchmarks becomes increasingly expensive (e.g., HLE's 1000 experts from 500+ institutions (Phan et al., 2025)). By transforming existing QA tasks into more challenging debate scenarios, we extend the useful lifespan of established datasets while raising the evaluation ceiling.

Importantly, our approach demonstrates **ASI readiness** by providing a theoretically unbounded measurement space. Unlike conventional benchmarks with fixed performance ceilings, debate assessments scale with model capabilities. This scalability operates on multiple fronts: the methodology can be seamlessly applied to any QA dataset, including the most difficult existing benchmarks and those yet to be created for future systems. The performance ceiling is also practically limitless; achieving a perfect score requires not merely correctness, but comprehensively out-arguing all opponents on all questions—an exceptionally high bar. The foundation of this future-proof design lies in its ability to facilitate weak-to-strong evaluation through structured debate. Our empirical results strongly ground this theoretical soundness: we demonstrated that even significantly weaker or contaminated models can reliably adjudicate debates and produce consistent rankings of stronger debaters (Table 3), corroborating findings from Khan et al. (2024). This unique combination of adaptability, a high performance ceiling, and empirically validated weak-to-strong assessment makes our framework a robust and valuable tool for evaluating advanced systems as they approach superintelligence.

The **cost efficiency** of our method represents a strategic trade-off. While evaluating a new model is more computationally intensive than standard few-shot methods, this is mitigated by a one-time benchmark creation cost and efficient partial tournaments for subsequent models. Critically, this approach becomes increasingly favorable over time: inference costs are predictably declining, while the expense of creating new, sufficiently hard benchmarks is escalating sharply (e.g., FrontierMath's mobilization of 60+ mathematicians (Glazer et al., 2024)). By recycling existing high-quality datasets, our framework offers a sustainable and future-proof evaluation path, sidestepping the prohibitive costs of perpetual benchmark creation while providing a high ceiling for even the most advanced models.

**Limitations**: Our study primarily employed only 50 MMLU-Pro questions, limiting domain-specific analysis reliability. While our complete evaluation of 5,500 debates generated over 11,000 rounds of argumentation across 11 models, the computational demands remain substantial—though justified when compared to creating entirely new benchmarks. Judge model biases, particularly toward persuasiveness rather than correctness, warrant further investigation (Liu et al., 2024). The applicability to QA-formatted visual benchmarks for Vision-Language Models (VLMs) also requires further study, especially given VLM-specific challenges like data contamination (Chen et al., 2024) and adversarial robustness (Dong et al., 2025a;b). Finally, the debate format's fixed structure (2-5 rounds) may not fully capture the nuances of complex reasoning, suggesting room for format optimization in future work.

In summary, our debate-driven approach offers a reproducible, contamination-resistant method for evaluating advanced language models, demonstrating that "pretraining on the test set is no longer all you need" for meaningful AI assessment.

## Ethics Statement

Our debate-driven evaluation framework introduces several design decisions that warrant ethical consideration. We identify and address these considerations below:

**Adversarial Role Assignment.**   Our framework intentionally assigns a "Con" model to argue against objectively correct answers. While this might initially appear to reward factual incorrectness, this adversarial approach serves a critical evaluation purpose: it forces deeper reasoning from both models, exposing superficial memorization. By removing multiple-choice distractors and allowing the Con model to construct its own alternatives, we mitigate the ethical concern of forcing models to defend demonstrably false positions. This approach aligns with common academic practices such as debate competitions and devil's advocate exercises, where arguing opposing viewpoints develops critical reasoning skills.

**Persuasiveness vs. Correctness.**   A potential concern with debate-based evaluation is that it might reward persuasiveness over factual correctness. We acknowledge this risk but find it diminishes significantly as question difficulty increases. For complex questions (e.g., in specialized domains like mathematics, physics, or medicine), persuasive rhetoric without substantive correctness becomes ineffective—similar to how academic evaluation prioritizes sound reasoning over presentation style. Our experiments demonstrate that judge models consistently favor debaters with stronger reasoning capabilities, and top-performing models in debates align well with traditional QA accuracy.

**Judge Model Bias.**   We recognize that LLM-based judges may inherit biases present in their training data. Our research mitigates this through multiple safeguards: (1) using structured QA with known correct answers rather than open-ended questions, (2) employing multiple judge models to verify consistency, (3) implementing double round-robin formats with role reversals to eliminate positional advantages, and (4) blind judging protocols that focus only on argument quality. These measures collectively reduce the impact of potential judge biases.

**Benchmark Accessibility.**   By publicly releasing our benchmark, detailed debate logs, and evaluation methodology, we promote transparency and reproducibility in AI evaluation. This open approach allows for community scrutiny and improvement of the framework while democratizing access to evaluation tools that would otherwise require substantial resources to develop.

We believe this framework represents an ethical advancement in AI evaluation that helps distinguish genuine reasoning from shallow pattern matching—a crucial capability as AI systems become increasingly integrated into high-stakes domains where real understanding, not mere memorization, is essential.

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

## A   Additional Debate Evaluation: Questioning-Only Heatmap

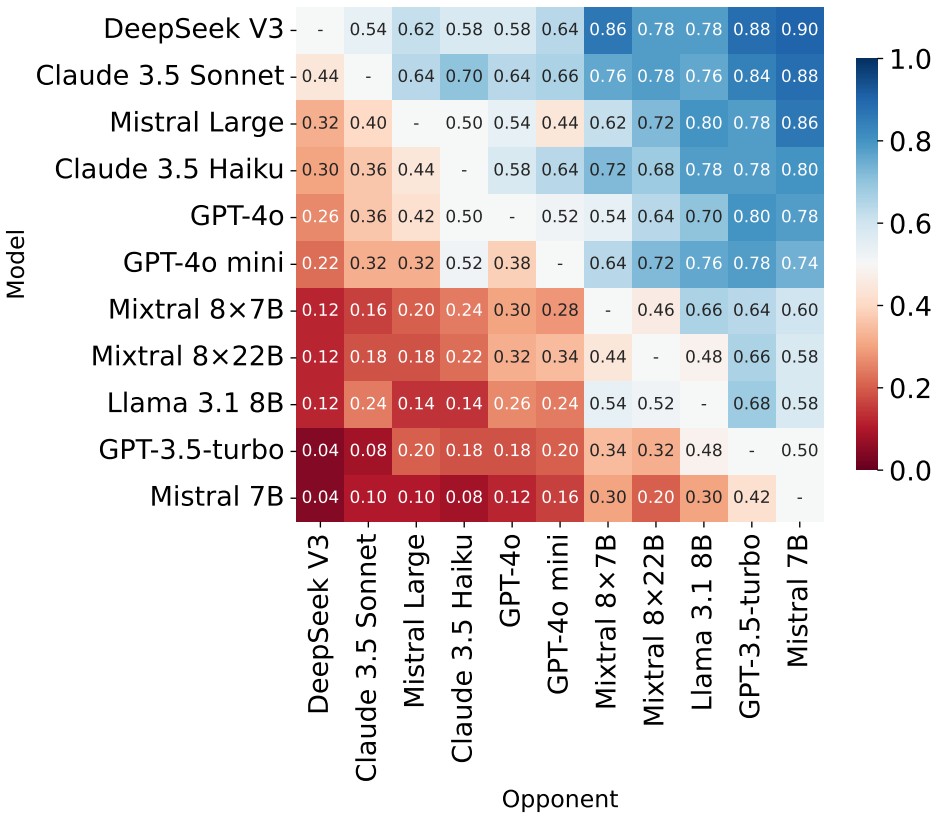

Figure 6: Head-to-head win rate heatmap (Questioning-Only). Each entry reflects win rates when models challenge the official answer. While noisier than defending-only results, the structure is broadly consistent.

## B   Judge Model Variation Results

Below are detailed win-rate tables showing pairwise comparisons between models across different judge models.

Table 4: Judge Failure: Mistral 7B. The model failed to adhere to output format instructions, leading to default judgments and the indiscriminate win rates shown.

| Opponent → | GPT-4o Mini | Mistral Large | Mistral 8×7B | Mistral 7B |
|---|---|---|---|---|
| GPT-4o Mini | – | 0.50 | 0.50 | 0.51 |
| Mistral Large | 0.50 | – | 0.50 | 0.50 |
| Mistral 8×7B | 0.50 | 0.50 | – | 0.50 |
| Mistral 7B | 0.49 | 0.50 | 0.50 | – |

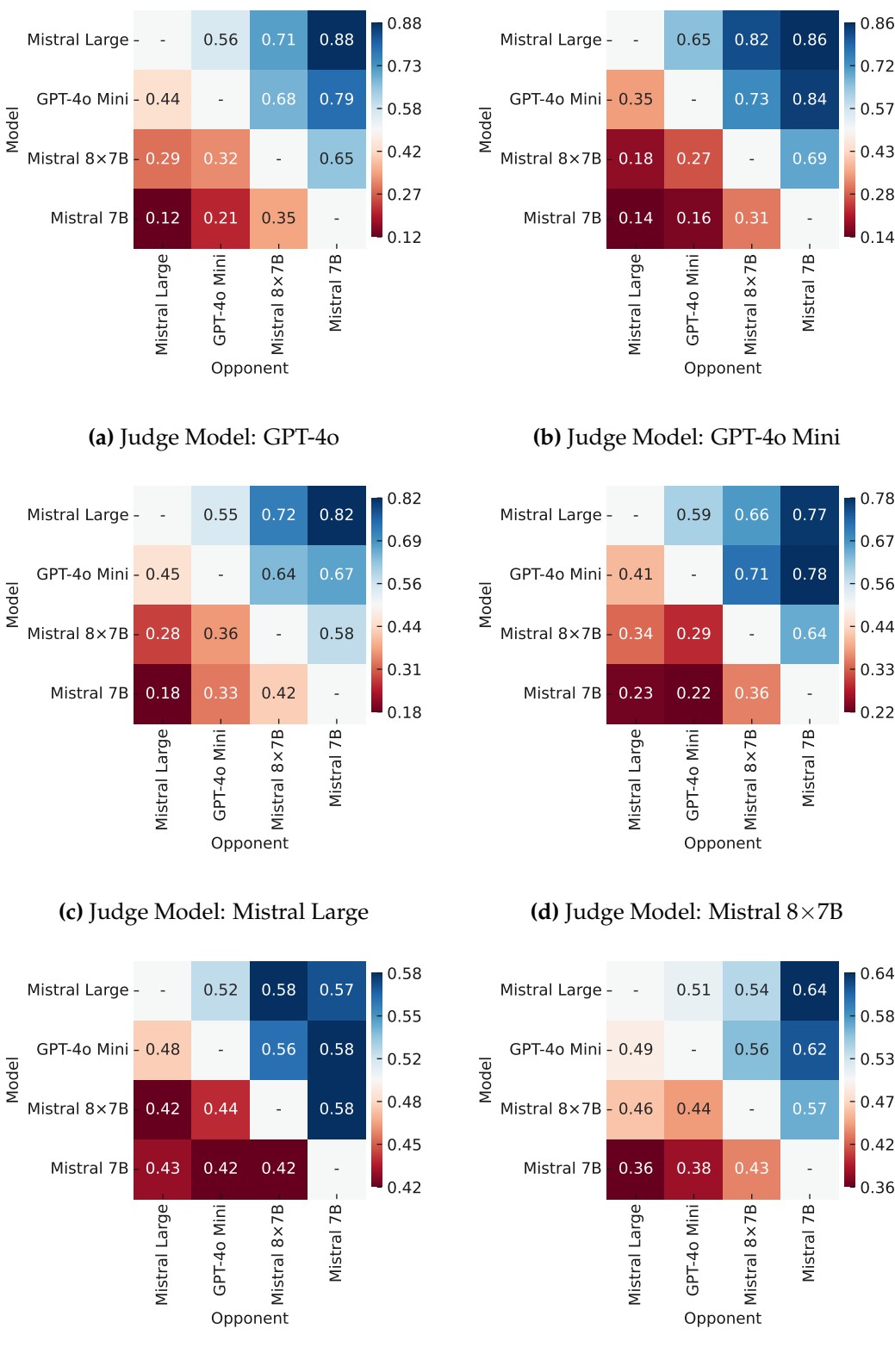

Figure 7: Pairwise win rate heatmaps for each judge model. Each subfigure (a)–(f) shows the win rates between models, as judged by the corresponding model.

## C  Confirmatory Evaluation on the GPQA Dataset

To confirm the generalizability of our framework beyond the MMLU-Pro dataset, we conducted a full evaluation on the GPQA main test set (Rein et al., 2023), which consists of 448 graduate-level questions. This experiment involved a round-robin tournament with five open-source models: *Llama 4 Scout* (Meta AI, 2025), *Llama 3.1 8B*, *Mixtral 8×7B*, *Mistral 7B*, and *Phi-4 Multimodal* (Microsoft, 2025).

The debates followed the same double round-robin protocol as the main experiment, with *Llama-4 Scout* serving as the judge model. The aggregated pairwise win rates are presented in Table 5. The results demonstrate a clear and consistent performance hierarchy with no intransitive loops (e.g., A > B > C > A), reinforcing the robustness and reliability of our debate-driven evaluation method on a different, challenging benchmark.

Table 5: Pairwise win rates on the GPQA main test set. Each cell $(i, j)$ shows the win rate of model $i$ against model $j$. The judge model was Llama 4 Scout.

| Opponent → | Llama 4 Scout | Llama 3.1 8B | Mixtral 8×7B | Mistral 7B | Phi-4 MM |
|---|---|---|---|---|---|
| Llama 4 Scout | – | 0.78 | 0.83 | 0.80 | 0.87 |
| Llama 3.1 8B | 0.22 | – | 0.51 | 0.61 | 0.83 |
| Mixtral 8×7B | 0.17 | 0.49 | – | 0.56 | 0.82 |
| Mistral 7B | 0.20 | 0.39 | 0.44 | – | 0.71 |
| Phi-4 Multimodal | 0.13 | 0.17 | 0.18 | 0.29 | – |

## D  Detailed TrueSkill and Bradley–Terry Scores

This appendix provides the full *TrueSkill* and *Bradley–Terry (BT)* rating tables illustrating how model scores change before and after introducing a fine-tuned *Llama 3.1 8B* model into the reference set of eleven models:

- **DeepSeek V3**
- **Claude 3.5 Sonnet**
- **Mistral Large**
- **Claude 3.5 Haiku**
- **GPT-4o**
- **GPT-4o mini**
- **Mixtral 8×7B**
- **Mixtral 8×22B**
- **Llama 3.1 8B**
- **GPT-3.5-turbo**
- **Mistral 7B**

All references below use *GPT-4o* as the judge model during debates.

From Tables 6 and 7, *TrueSkill* scores exhibit minimal shifts among reference models, whereas *Bradley–Terry* adjustments can be substantially larger. This highlights the greater stability of TrueSkill for incremental or partial-tournament settings.

Table 6: TrueSkill scores comparing reference-only results to updated ratings after adding the fine-tuned *Llama 3.1 8B*.

| Model | Score (ref models) | Score (with new model) |
|---|---|---|
| DeepSeek V3 | 492.00 | 489.40 |
| Claude 3.5 Sonnet | 468.10 | 468.20 |
| Mistral Large | 447.80 | 447.90 |
| Claude 3.5 Haiku | 436.20 | 436.30 |
| GPT-4o | 414.50 | 414.90 |
| GPT-4o mini | 411.60 | 412.00 |
| Mixtral 8×22B | 341.30 | 341.40 |
| Mixtral 8×7B | 346.50 | 346.60 |
| Llama 3.1 8B | 326.30 | 326.50 |
| GPT-3.5-turbo | 289.70 | 290.10 |
| Mistral 7B | 281.70 | 281.80 |
| *Llama 3.1 8B Finetuned* | 0.00 | <315.20> |

Table 7: Bradley–Terry scores comparing reference-only results to updated ratings after adding the fine-tuned *Llama 3.1 8B*.

| Model | Score (ref models) | Score (with new model) |
|---|---|---|
| DeepSeek V3 | 1963.00 | 1981.00 |
| Claude 3.5 Sonnet | 1881.00 | 1904.00 |
| Mistral Large | 1731.00 | 1755.00 |
| Claude 3.5 Haiku | 1705.00 | 1728.00 |
| GPT-4o | 1637.00 | 1660.00 |
| GPT-4o mini | 1608.00 | 1632.00 |
| Mixtral 8×22B | 1308.00 | 1331.00 |
| Mixtral 8×7B | 1325.00 | 1349.00 |
| Llama 3.1 8B | 1243.00 | 1271.00 |
| GPT-3.5-turbo | 1083.00 | 1107.00 |
| Mistral 7B | 1015.00 | 1039.00 |
| *Llama 3.1 8B Finetuned* | 0.00 | <1244.00> |

# E  Debate Prompts

This appendix provides the exact prompts used to structure debates between models in our evaluation framework.

## E.1  Positive (Pro) Side Prompt

The model assigned to defend the correct answer receives the following prompt:

```
You are the POSITIVE side. Argue in favor of the correct answer.
Consider and address any previous counter-arguments:
Question: {question_text}
Your answer: {correct_answer}
Debate history:
{debate_transcript}
```

## E.2  Negative (Con) Side Prompt

The model assigned to challenge the correct answer receives this prompt:

```
You are the NEGATIVE side. The system has rejected the answer as
incorrect. Provide an alternative answer and argue against the
original answer, considering previous arguments:
Question: {question_text}
Claimed correct answer: {correct_answer}
Debate history:
{debate_transcript}
```

### E.3 Judge Prompt

The judge model, which evaluates debates without knowing the official answer, receives this prompt:

```
You are an impartial judge evaluating a debate about the answer to
a question. Based solely on the quality of arguments presented by both
sides, determine which side has made the stronger case.

The debate transcript is provided below. You do not need to determine
the correct answer yourself - only which debater presented more
convincing arguments.

{debate_transcript}

After careful consideration of all arguments, which side made the stronger
case overall? Respond with only "positive", "negative", or "continue"
(if more debate rounds are needed for a clear determination).
```

