# OpenReview forum: "Pretraining on the Test Set Is No Longer All You Need: A Debate-Driven Approach to QA Benchmarks"
_colmweb.org/COLM/2025/Conference — COLM 2025_

### Official Review · Reviewer_5Sib · 2025-05-06

**Rating:** 6
**Confidence:** 3
**Ethics Flag:** 1

**Summary:**

This paper presents an innovative evaluation method to address potential benchmark contamination issues. Instead of relying on standard benchmarks, it introduces multi-round arguments between models to assess whether a model truly understands the content. I find this approach intuitive, as allowing a model to elaborate on facts makes it easier to evaluate its mastery. My detailed comments are below.

**Reasons To Accept:**

I believe this evaluation setup is straightforward and effective. The empirical results show that this method successfully differentiates genuine reasoning from memorization, as evidenced by the fine-tuned model's reduced debate performance despite improved QA accuracy. It also proves robust across various judge models.

**Reasons To Reject:**

The primary concern with this method is that *evaluating data points is too expensive*. Assessing a single data point requires agent-style back-and-forth and using an LLM as a judge. Could you provide the time required to evaluate 1,000 data points with a specified length? I’m open to inference frameworks like vLLM or SGLang, but since this involves multi-turn debates, I suspect the speed gains may be limited.

Additionally, based on Table 1, the difference between debatable and simple evaluations isn’t significant. I think most would prefer evaluating across diverse benchmarks to demonstrate their model’s general capabilities rather than focusing on debatable evaluations for a single dataset. I’d appreciate a comparison of time and cost, especially since you’re using an LLM-as-a-judge approach.

There are also some reference I thought may be beneficial to include:
- [1] Proving test set contamination in black-box language models.
- [2] Investigating Data Contamination in Modern Benchmarks for Large Language Models.
- [3] Detecting Pretraining Data from Large Language Models.

---

> ### Author Response · Authors · 2025-05-31
>
> Thank you for your insightful feedback. We address your concerns below:
>
> **1. Evaluation cost considerations:** Our vision centers on addressing the inevitable trajectory of AI evaluation: as frontier models advance exponentially, constructing benchmarks that meaningfully challenge them requires ever-increasing resources. Consider FrontierMath, which already demands substantial expert involvement. At some point, the expense of developing new, sufficiently difficult datasets will become unsustainable. Meanwhile, inference costs continue falling with GPU improvements and optimization techniques. Our method sidesteps this escalating cost entirely—dataset construction expense is effectively zero, while the framework remains capable of assessing arbitrarily strong models since weaker judges can distinguish superior reasoning in debates (Khan et al., 2024). Our new GPQA experiments further validate this approach.
>
> Addressing your specific inquiry about costs: our complete 11-model tournament required approximately $900 using commercial APIs, though this figure represents the one-time benchmark creation expense. For evaluating a new model, costs are dramatically lower due to our partial tournament approach. The complete absence of transitivity violations allows us to reduce evaluation from O(n²) full tournaments to O(log n) targeted matches, where n represents reference model count. This translates to ~6.7 - 18.8x the expense of traditional 5-shot evaluation—a modest increase compared to commissioning entirely new expert-crafted benchmarks. **Regarding timing specifics, exact figures depend heavily on hardware configurations and API rate limits; local deployment via vLLM remains feasible but would require optimizing concurrent model hosting.**
>
> **2. Performance gap between debate and simple evaluation:** We appreciate this observation. **Table 1's interpretation is admittedly challenging—we cannot know each model's contamination level, making direct comparisons difficult.** What we do demonstrate is contamination resistance: our Llama 3.1 fine-tuning experiment (50%→82% QA accuracy, but worse debate performance) provides concrete evidence that debate evaluation catches what simple evaluation misses. Contamination concerns are already paramount in the community, with solutions like LiveBench emerging. Our method offers contamination-free evaluation from day one and remains unsaturated indefinitely. Data contamination threatens diverse benchmarks equally—most traditional QA benchmarks risk contamination despite their quality, except confidential or dynamic methods like SWE-Bench that focus on specific domains. Our work re-enables assessment of models on general QA datasets already at risk, providing contamination-free insights while extending these datasets' lifespan. The method remains inherently future-proof because: (A) weaker judges can effectively evaluate stronger models in structured debates, and (B) the theoretical performance ceiling extends far beyond simple accuracy—models must excel at both defending correct answers AND constructing plausible alternatives across all opponents, creating a much higher bar for true mastery.
>
> **3. Recommended citations:** Thank you for these excellent suggestions! We will incorporate these papers to strengthen our narrative on contamination detection.
>
> **References:**
>
> - [Debating with More Persuasive LLMs Leads to More Truthful Answers](https://openreview.net/forum?id=iLCZtl7FTa) (Khan et al., 2024)

---

> > ### Comment · Reviewer_5Sib · 2025-06-04
> > **Thank you for your response**
> >
> > Dear Authors,
> >
> > Thank you for your detailed response. I greatly appreciate the time and effort you put into addressing my concerns in your rebuttal. To be honest, I’m still somewhat hesitant about the final assessment. My primary concerns aren’t with your method itself but with the practical application of dynamic evaluations in this field. Given that over ten works have approached this topic from various angles, none seem to be widely adopted in practice, which makes me question the real-world applicability of these methods.
> >
> > I’ve reviewed your additional experiments on GPQA, and these results are encouraging. They suggest that your approach could potentially convince other researchers to adopt these methods, helping to address the ongoing challenge of decontamination in research. Based on this progress, I’m inclined to increase my score. Thanks!

---

> > > ### Author Response · Authors · 2025-06-05
> > >
> > > Thank you for your candid feedback—we really appreciate your honesty!
> > >
> > > Our method's fundamental difference is that it isn't just another dynamic evaluation approach. Think of it like this: we offer a pipeline for evaluation upgrade on any QA dataset—"debate-enhanced, standardized with a model group" is analogous to "few-shot vs 0-shot" or "CoT vs non-CoT" in traditional evaluation pipelines. We evaluate on the same dataset but add fixed reference models to measure domain expertise through debate performance, offering domain-specific and contamination-free insights.
> > >
> > > Regarding the adoption concerns about dynamic evaluation methods—it’s right that many suffer from reproducibility, comparability, and efficiency challenges. We tackle reproducibility and efficiency by pre-constructing debate logs through full round-robin tournaments. We also anticipate that adoption of dynamic evaluation methods will grow as model capabilities advance. For example, back in the Claude 3 era, Anthropic advertised their model using traditional benchmarks like MMLU, GPQA, and GSM8K. But with Claude 4, they're advertising with various agentic benchmarks and even proactively demonstrate model capabilities by "playing Pokémon"—hinting that the transformation might have already begun.
> > >
> > > Notably, new models like Claude 4 rarely use MMLU and similar datasets, partially due to contamination concerns. Our method re-enables reasonable evaluation on these test sets—even if models have been contaminated (unintentionally or otherwise), and erase public concerns about potential contamination. This vastly improves the continued relevance and recognition of established benchmarks.
> > >
> > > ---
> > >
> > > Thanks again for engaging so thoughtfully with our work—your perspective on practical adoption is valuable and helps us better position our contribution!

---

### Official Review · Reviewer_fs1s · 2025-05-11

**Rating:** 5
**Confidence:** 3
**Ethics Flag:** 1

**Summary:**

The paper proposed a debate-driven evaluation paradigm that transforms an existing QA dataset into structured adversarial debates, where one model is given the official answer to defend and another constructs and defends an alternative answer. Empirical results validated the robustness of the method and its effectiveness against data contamination.

**Questions To Authors:**

1. Figures 1, 5-7  were not referred to in the body of the paper. Tables 9 and 10 in Sec. 5.6 do not exist in the body but are in the Appendix.
2. For the paragraph, Partial Tournament Validity, no results (evidence) seem shown.

**Reasons To Accept:**

The paper proposed a debate-driven evaluation paradigm that transforms an existing QA dataset into structured adversarial debates. Empirical results validated the robustness of the method and its effectiveness against data contamination.

**Reasons To Reject:**

1. While the authors claim their approach deals with any existing QA dataset, the experiments were done only for a dataset, MMLU-Pro. Further, the size of the dataset for evaluation, 50 randomly sampled questions, is rather small. I wonder it can yield a reliable result.
Most importantly, the authors should investigate and discuss how large dataset should be involved for the debate to yield reliable judgments.
2. While the authors claim retaining only questions and correct answers, and removing incorrect alternatives are better, it was not verified with any evidence. So it is better to show whether using the original incorrect answers in the debate is really harmful in the experiments.
3. It was not clearly discussed why the rounds of the debate should be in the range of 2-5.
4. While it costs less than constructing a new dataset, the proposed approach seems rather costly as an evaluation method. Thus, the authors should discuss its computational cost and compare it with other related approaches.

---

> ### Author Response · Authors · 2025-05-31
>
> Thank you for your constructive feedback. We address each concern below:
>
> **1. Sample size and dataset diversity:** We acknowledge this concern, which we noted in our limitations. To address it directly, we conducted additional large-scale evaluation on the complete GPQA dataset (448 questions)—a completely different benchmark from MMLU-Pro—using 6 open-source models. Results strongly validated our approach's generalization: no transitivity violations emerged in overall rankings. Notably, achieving clean results with just 50 questions actually demonstrates the method's effectiveness. We agree that investigating optimal dataset sizes and question-type impacts represents valuable future work.
>
> **2. Removing incorrect options:** This design choice addresses ethical alignment rather than evaluation effectiveness. We found it inappropriate to reward models for defending objectively incorrect answers, as this could encourage deceptive reasoning. By allowing Con models to construct alternatives, we test genuine reasoning within reasonable constraints.
>
> **3. Round selection (2-5):** Our range aligns with established literature. Liang et al. (2024) studied 1-4 rounds while Du et al. (2024) examined 1-5 rounds, both finding diminishing returns beyond this scope. Our minimum of 2 rounds prevents judge contamination effects while maintaining efficiency, as additional rounds significantly increase computational costs.
>
> **4. Computational cost:** We share this concern about evaluation efficiency. Our approach targets long-term sustainability: as LLMs rapidly advance, creating sufficiently challenging datasets becomes increasingly expensive (e.g., FrontierMath already requires substantial expert involvement). Crucially, benchmark creation costs trend upward while inference costs decline with hardware advances—making our compute-intensive approach increasingly practical over time. Eventually, dataset creation costs may become prohibitive. Our method offers zero dataset construction cost while leveraging Khan et al.'s (2024) finding that weaker judges can reliably evaluate stronger models. The partial tournament property (no transitivity violations) enables O(log n) evaluation complexity versus O(n²), where n is the number of reference models, yielding only ~6.7 - 18.8x more cost compared to standard 5-shot evaluation.
>
> **Questions:** Thank you for identifying inconsistencies in formatting—these will be corrected. Regarding partial tournament validity: Figure 6's complete absence of transitivity violations provides empirical evidence, enabling binary-search-like evaluation of new models. TrueSkill scoring ensures comparable benchmark scores across evaluations.
>
> **References:**
>
> - [Encouraging Divergent Thinking in Large Language Models through Multi-Agent Debate](https://aclanthology.org/2024.emnlp-main.992) (Liang et al., 2024)
> - [Improving Factuality and Reasoning in Language Models through Multiagent Debate](https://arxiv.org/abs/2305.14325) (Du et al., 2024)
> - [Debating with More Persuasive LLMs Leads to More Truthful Answers](https://openreview.net/forum?id=iLCZtl7FTa) (Khan et al., 2024)

---

> > ### Comment · Reviewer_fs1s · 2025-06-02
> >
> > Thank you for the additional results and the detailed responses. I will raise my score.

---

> > > ### Author Response · Authors · 2025-06-02
> > >
> > > Thank you for taking the time to review our additional experiments and detailed responses. We greatly appreciate your willingness to reconsider and for raising your score.
> > >
> > > If you have any remaining questions or concerns about our work, we would be happy to address them. Your constructive feedback has been invaluable in strengthening our paper.

---

### Official Review · Reviewer_SpHc · 2025-05-12

**Rating:** 7
**Confidence:** 3
**Ethics Flag:** 1

**Summary:**

This paper introduces a debate-driven approach to evaluate the reasoning capabilities of large language models (LLMs). Current evaluation benchmarks face two major challenges: the increasing cost of developing more complex benchmarks as models advance, and the issue of data contamination, where benchmark questions may overlap with training data. These limitations highlight the need for a new evaluation framework. In response, the authors propose a novel method that assesses the validity of answers to existing multiple-choice question-answering (MCQA) samples through structured debate. The effectiveness of this framework is demonstrated through experimental validation.

**Questions To Authors:**

(L197) Since exact match requires instruction following ability, differences between models are expected to emerge. Is this a valid measurement method for traditional QA accuracy?

**Reasons To Accept:**

- The motivation is well understood and important for the LLM community.
- The experimental results (Figure 7) are consistent and appear to demonstrate the effectiveness of the evaluation method.
- Fine-tuning impact assessment is an interesting finding.

**Reasons To Reject:**

- Not only the cost of constructing the benchmark set, but also the cost of evaluation is an important issue, and we would like to see an explicit mention of how much computing cost is required.

- (L.232) Since robustness is a very important issue for evaluation frameworks, it would be helpful to have a more detailed explanation to determine whether Partial Tournament Validity has been sufficiently evaluated.
The computing cost does not seem to be significantly different from other experiments, so it would be useful to be able to evaluate under different conditions if necessary.

- Although it is understood that the purpose is not to create a model leaderboard, still, an explanation is desired why these 11 models were selected.

---

> ### Author Response · Authors · 2025-05-31
>
> Thank you for your positive assessment and thoughtful questions! We're happy to provide clarifications:
>
> **1. Computational cost:** You raise an important practical concern. Indeed, our method requires ~6.7 - 18.8x more computation than standard 5-shot evaluation for our created benchmark. However, our framework targets a fundamentally different challenge: as we've witnessed the progression from GPT-3 to GPT-4 to o1 in just a handful of years, LLM capabilities are advancing exponentially. Eventually, creating sufficiently challenging benchmarks will become prohibitively expensive or even impossible. This trend diverges sharply from inference costs, which decrease predictably with hardware advances. Our approach offers sustainable evaluation at a fraction of future benchmark creation costs, while remaining effective as models grow stronger—since weaker judges can reliably evaluate superior reasoning in structured debates. Moreover, our absolute ceiling is exceptionally high—achieving perfect scores requires a model to comprehensively outperform all other participants in both defending correct answers and constructing plausible alternatives.
>
> **2. Partial tournament validity:** Thank you for noting this—we should clarify that this isn't an assumption but rather an empirical finding emerging from the remarkable property of extreme transitivity. This exceptional result enables us to confidently predict: if Model A > Model B and a new model beats A, it will beat B; if it loses to B, it will lose to A; if in between, we have a precise score range. While practical precision is limited to placement between two adjacent models, more granular rankings cannot be achieved. We consistently find extreme transitivity across all experiments: the original 11-model MMLU-Pro test, the different judge test, and the new full GPQA evaluation—dramatically reducing requirements from O(n²) to O(log n) (from full round-robin to debating with existing models in a binary search-like method).
>
> **3. Model selection rationale:** The 11 models were thoughtfully selected to ensure comprehensive validation. We covered: strong commercial models, strong open-source alternatives, weak commercial options, and weak open-source baselines. This diversity spans major model providers, both open and closed source, frontier capabilities and historical references. Eleven models provide robust validation, with additional models easily incorporated after benchmark publication.
>
> **4. Evaluation pipeline clarification:** Apologies for the confusion. For traditional QA accuracy (Line 197), we didn't create a custom evaluation pipeline but directly reused MMLU-Pro's test pipeline with our filtered data. Their original method employs regex pattern matching to extract letter choices with exact comparison, ensuring our baseline measurements align with established practices.

---

> > ### Comment · Reviewer_SpHc · 2025-06-07
> > **Thank you for your response**
> >
> > Thank you for your response to my questions and for providing the additional experimental results on GPQA.
> >
> > While I believe it may still be premature to conclude that the proposed debate-driven approach can serve as a viable alternative to existing benchmark-based evaluation methods for addressing the benchmark saturation problem, I appreciate the work as an important step toward exploring a promising new direction in this area of research.

---

> > > ### Author Response · Authors · 2025-06-09
> > >
> > > Thank you for your thoughtful perspective and encouraging words. We couldn't agree more with your assessment. You've perfectly captured our view by describing this work as an "important step" and a "promising new direction"—it is precisely this foundational contribution we aimed to make.
> > >
> > > While its role as a full "viable alternative" is indeed something the future will determine, we are incredibly encouraged by the framework's present-day robustness. The consistent, transitive rankings we observed—not only on our initial MMLU-Pro sample, but across the entire GPQA dataset and a diverse suite of both open-source and proprietary models—provide a solid empirical bedrock for this new direction.
> > >
> > > Our hope is that this work serves as a launchpad, inspiring the community to build upon this approach as we collectively navigate the future of AI evaluation. We are deeply grateful for your support and for recognizing the potential of this research.

---

### Official Review · Reviewer_wZxD · 2025-05-12

**Rating:** 7
**Confidence:** 4
**Ethics Flag:** 1

**Summary:**

This paper proposes a debate-like evaluation method that addresses benchmark saturation and data contamination. The paper is clearly motivated, presents a well-structured and reproducible pipeline for converting existing benchmarks into a debate-style format, and concludes with sound reasoning and empirical evidence on the cost-effectiveness and contamination robustness of the proposed approach. Although debate-based evaluation has been explored before, the structured and standardized format proposed here is novel and shows future potential.

**Questions To Authors:**

**Q1:** Can your method work with tasks other than QA?

**Q2:** The paper does not mention how the 50 MMLU-Pro questions were sampled. Since MMLU-Pro spans a wide range of domains, domain selection could significantly affect debate outcomes--especially if some domains favor factual recall while others favor reasoning. Could the authors clarify the sampling methodology and the domain distribution of the 50 selected questions?

**Recommended Citations**

There are recent methods from the computational linguistics (CL) community that are similar in spirit. I recommend the authors review and cite the following:

- [QUDeval: The Evaluation of Questions Under Discussion Discourse Parsing](https://aclanthology.org/2023.emnlp-main.325) from  EMNLP 2023
- [Discursive Socratic Questioning: Evaluating the Faithfulness of Language Models' Understanding of Discourse Relations](https://aclanthology.org/2024.acl-long.341/) from ACL 2024

Both works use multiple question answering (similar to your debate approach) to evaluate how well models understand context.

Update on June 16th: Increased my score slightly after author response.

**Reasons To Accept:**

**S1: The presentation of the paper is good.**
The paper begins by clearly presenting benchmark saturation and data contamination as its motivation. In the conclusion, it clearly summarizes how the proposed method addresses these two problems. The writing style is coherent and easy to follow.

**S2: The evaluation appears to be OK.**

- I especially enjoyed the visualization in Figure 7 (although Figures 5 and 6 should be renamed as Figure 7a and 7b).
- The results in Section 5.4 provide strong evidence for the paper's claim: improvement on the standard benchmark by fine-tuned models does not translate to improvement on the proposed debate benchmark.

**Reasons To Reject:**

**W1: Correlation with existing benchmarks/human judgments.**
My primary criticism is that, although the paper shows that improvement on standard benchmarks does not translate to the proposed evaluation, it does not clearly demonstrate whether the evaluation correlates with existing benchmarks or human judgments. In other words, while the paper shows that the proposed evaluation can disentangle data contamination, its reliability remains unclear due to the lack of grounding in established methods. Can the authors explore other works on automatic model evaluation and report correlation? (Please correct me if I missed this in the paper).\
I think a good or moderate correlation with existing metrics would be sufficient, it would show that the proposed evaluation shares common ground with established metrics while also highlighting where it diverges.

**W2: The soundness of evaluation can still be improved in the following aspects:**

- Many MMLU-Pro questions have a specific correct answer (e.g., math problems). Forcing a model to argue for an incorrect answer may place it at a disadvantage, as it could be more difficult to find supporting evidence for an incorrect position. Although role-switching ensures both sides are disadvantaged equally, alternative answers are not fixed, and defending different incorrect answers may vary in difficulty. This undermines the fairness assumed by role-switching.
- Figure 3 is somewhat vague regarding model pairing. Are models first paired in a round-robin format with a random question assigned to each match, or is a complete round-robin tournament played for each question?
- In some evaluations, the same LLM is both contestant and judge. For example, Table 1 shows GPT-4o acting as both judge and contestant. Could this be unfair, as the model may favor its own chain of reasoning?
- The paper does not provide judging rubrics to the judge LLM. The concept of "making the better case" is entirely subject to the judge model's interpretation. This could affect the fairness of the benchmark if different judges are used. It may be helpful to standardize the judge (or use an ensemble) to mitigate such bias.
- While the paper thoroughly studies the impact of contamination on contestant models, it does not investigate contamination effects on judge models. Although the paper tries to make the judge models answer-blind by not revealing answers, they may still have encountered the questions and answers during training. It would be worth testing whether such prior exposure affects the fairness of judgments.

---

> ### Author Response · Authors · 2025-05-31
>
> Thank you for your thoughtful attention to our work and valuable feedback!
>
> **W1. Correlation with existing benchmarks/human judgments:** We address this multifaceted concern from several angles:
>
> *Data contamination validation:* Our new experiments with fine-tuned models empirically demonstrate contamination detection capabilities. Additionally, our full GPQA evaluation strengthens these findings.
>
> *Correlation with MMLU-Pro:* We computed Pearson correlation between QA accuracy and debate wins on our 50-question sample across 11 models, finding r=0.90 (p=0.00017). However, we view this correlation cautiously since contamination can artificially inflate it—a contaminated weak model could easily top traditional benchmarks while failing in debates.
>
> *Independence as a feature:* We offer a fundamentally new evaluation paradigm that generally aligns with model capabilities (stronger models like GPT-4o clearly outperform GPT-3.5/Mistral-7B) while providing unique insights. Correlating with diverse datasets may be less informative—divergences would be difficult to interpret given dataset heterogeneity. Regarding human correlation, extensive literature (Auto-Arena; Zhao et al., 2024, and LLM-as-Judge surveys; Li et al., 2024; Gu et al., 2024) establishes LLM judging capabilities. While human validation would strengthen our claims, recruiting domain experts across multiple fields remains prohibitively expensive.
>
> **W2. Evaluation soundness concerns:**
>
> *Alternative vs. incorrect answers:* Ultimately, we test LLMs' domain expertise—in the Pro role, this manifests as defending correct answers with sound reasoning; in the Con role, it translates to constructing plausible alternatives under constraints and justifying them convincingly. With the official answer removed, the solution space becomes vast, framing this as an engineering problem requiring deep domain knowledge. We consider rewarding defense of objectively incorrect choices ethically problematic. Our role-switching protocol mitigates positional bias, as evidenced by aligned rankings across overall, defend-only, and question-only scenarios (Figure 4), demonstrating we assess fundamental domain proficiency.
>
> *Tournament structure:* To clarify: we conducted complete round-robin tournaments for each of 50 questions across all 55 model pairs, with role-switching within each pairing.
>
> *Self-judging concerns:* While literature documents this theoretical concern, our empirical analysis proves reassuring. All judges produced aligned results despite some models serving dual roles. Interestingly, we observed both cases where models scored themselves lower (Mistral Large received fewer wins when judging itself compared to other judges) and higher (GPT-4o Mini, Mixtral-8×7B) relative to their scores under different judges. These minimal variations don't compromise validity or alter rankings(i.e., aligned overall rankings, zero internal loops in all judges).
>
> *Judging rubrics:* Your concern is valid. Our empirical findings—consistent results across different judges with complete transitivity (no loops)—suggest we're measuring something genuine and robust. This consistency emerges despite varied judge interpretations.
>
> *Judge contamination:* Excellent suggestion! We immediately conducted follow-up experiments with contaminated judges (Llama-3.1-8B fine-tuned on test data). Results showed contamination has subtle effects on scoring, but rankings remained consistent with complete transitivity preserved. This suggests our role-switching design successfully neutralizes potential biases—both debaters are equally affected, maintaining fair evaluation.
>
> **Q1. Extension beyond QA:** Currently limited to QA tasks, though future adaptations are possible. QA provides crucial grounding through known correct answers—ensuring one debater defends factual content. This design forces Con models to demonstrate genuinely deeper understanding to persuade judges, significantly reducing bias compared to open-ended tasks where both sides might argue incorrect positions.
>
> **Q2. Sampling methodology:** We employed random sampling, resulting in coverage across all 14 MMLU-Pro domains with 2-6 questions per domain, ensuring representative evaluation.
>
> **Q3. Citations:** We greatly appreciate these recommendations and will incorporate them to strengthen our work!
>
> **References:**
>
> - [Auto-Arena: Automating LLM Evaluations with Agent Peer Battles and Committee Discussions](https://arxiv.org/abs/2405.20267) (Zhao et al., 2024)
> - [LLMs-as-Judges: A Comprehensive Survey on LLM-based Evaluation Methods](https://arxiv.org/abs/2412.05579) (Li et al., 2024)
> - [A Survey on LLM-as-a-Judge](https://arxiv.org/abs/2411.15594) (Gu et al., 2024)

---

> > ### Comment · Reviewer_wZxD · 2025-06-04
> > **Thanks for your response & New questions**
> >
> > Hi authors:
> >
> > Thank you for writing the detailed response.
> >
> > ## W1
> > As for Weakness 1 (W1), I understand your metric's relation with data contamination (which is a valid novelty). But what I asked in the initial review was to substantiate your metric as a reliable measure.
> >
> > My suggestion was to provide intrinsic and extrinsic validation to support the claim that your metric is reliable. Analyzing the correlation with human judgement will be most helpful, but the authors claimed it infeasible. However, I do believe there should be indicative human evaluations (even in a small size), because your paper is about judging the effectiveness of models.
> >
> > Correlation with other metrics (which is dataset agnostic) will also help. The paper now presents with QA accuracy (which shows a high correlation 0.9). I am not clear that if QA accuracy is the only metric for evaluating models. Can you tell us the standard practice in model evaluation and other possible metrics (and why you don't choose them for correlation comparison)?
> >
> > And most importantly, given the high correlation, I am also not clear what new insights are brought by the new metric. Can you please articulate?
> >
> > Can you also articulate how this 0.9 accuracy was derived? Is it instance-level correlation or system-level? Thanks!
> >
> > I am also worried about the claim that the evaluation is ''fundamentally new'' in your response. Since debate-based evaluation and multi-round evaluation are both well studied, I would tuned that down into a ''new intersection'' discovered by your team. :)
> >
> > Another questions was that I find the evaluation was only on 50 questions in MMLU-Pro. Can you show us the distribution of domains in these fifty questions? I am also worried that 50 is a too small size. Have you found the results are stable with this number? What would be the common practice for setting the evaluation size in model evaluation domain?
> >
> >
> > ## W2
> > As for Weakness 2 (W2), several responses have answered the questions in the initial reviews. But I still have reservations about the first point (i.e. the nature of incorrect answers, especially in math domain that incorrect answers can be too artificial).
> >
> > Can the authors provide case studies for incorrect answers for the domains that the paper has studied? I looked into the appendices but couldn't find any.
> >
> > This point also goes back to the most important question in W1. We are not yet clear about what new information will your proposed metric inform us beyond existing ones. Can you show a few instance level analysis to support the responses you may have for W1? Thanks!

---

> > > ### Author Response · Authors · 2025-06-10
> > >
> > > We're truly grateful for your detailed feedback and insightful follow-up questions—your thorough engagement has been incredibly valuable to us. Please let us know if there's anything at all we can clarify; we're eager to address any remaining concerns.

---

> > ### Author Response · Authors · 2025-06-05
> >
> > Thank you for these thoughtful follow-up questions! We appreciate the opportunity to clarify.
> >
> > **W1:**
> >
> > 1. Our method tests Pro models on defending correct answers and rewards Con models for constructing and defending alternatives—both positively correlated with domain expertise. The aligned rankings (question-only, defend-only, and overall) in Figure 4 demonstrate this nicely, with stronger models scoring higher across all three metrics.
> > 2. We agree human evaluation would be valuable! However, recruiting domain experts across 14 MMLU-Pro domains is quite costly, especially given the graduate-level expertise required.
> > To address this, we've made all debate transcripts publicly available in our supplementary materials (logs folder), containing debate histories for both main and judge analysis experiments. We welcome follow-up studies using this data!
> >
> > From our own inspection of familiar domains, we've seen compelling examples. In question 11329 (heat transfer coefficient), Claude-3.5-Sonnet defeated GPT-4o in both roles: as Pro, defending 1.08 Btu/hr-ft²-°F with detailed calculations and rigorous justification; as Con, constructing a plausible alternative (0.92) using different valid correlations (Churchill and Chu) and citing industry-standard ranges (0.8-1.0). This transparency helps mitigate concerns about human validation.
> >
> > 3. Regarding metrics: QA accuracy is indeed the most widely adopted evaluation method for models. The standard practice in MMLU-Pro uses 0-shot CoT accuracy (%), which is what we correlated with debate wins in Table 1—this is **system-level** correlation.
> >
> > Different evaluation protocols (e.g., human preference in Chatbot Arena using ELO scores, real-world capabilities in SWE-Bench) typically remain independent—that's why we focused on within-benchmark correlation, rather than explore more. The 0.9 correlation suggests debate performance generally aligns with original MMLU-Pro performance but captures subtle differences, offering contamination-free, domain-specific, model-group-specific insights.
> >
> > The "fundamentally new" aspect is that we provide a pipeline for evaluation upgrade on any QA dataset—think of "debate-enhanced, standardized with a model group" like "few-shot vs 0-shot" or "CoT vs non-CoT" in traditional evaluation pipelines. We evaluate on the same dataset but add fixed reference models to measure domain expertise through debate performance, yielding contamination-free insights.
> >
> > 4. Domain distribution in our 50 questions:
> >
> > Biology: 3, Business: 3, Chemistry: 5, Computer Science: 2, Economics: 3, Engineering: 4, Health: 3, History: 2, Law: 5, Math: 6, Philosophy: 2, Physics: 5, Psychology: 3, Other: 4
> >
> > We acknowledge 50 is a small sample, but our additional GPQA full dataset evaluation showed perfectly aligned patterns. This convergence from just 50 questions actually demonstrates the method's robustness!
> >
> > For dataset size, different goals require different scales: quick efficient evaluation might need only 50 questions (as we've shown), while comprehensive assessment would ideally use the full dataset. Our primary limitation is resources—without sponsorship, running 10,000+ tournaments across all models is challenging. We're excited about this method's future potential! :D
> >
> > ---
> >
> > **W2:**
> >
> > 1. To clarify: we study *alternative* answers, not incorrect ones, to avoid ethical dilemmas. How does this work in math debates? Looking at Claude-3.5-Sonnet vs GPT-3.5-Turbo: In Q7700, Con won by correcting Pro's double-counting error while reaching the same answer. In Q8005 (field extension), models constructed sophisticated alternatives arguing Q(√2×√3)=Q(√6) vs Q(√2,√3), demonstrating advanced algebraic understanding. Pro models also win by soundly defending correct answers. These patterns reveal mathematical expertise—models either defend through sound logic or construct mathematically plausible alternatives, both requiring genuine expertise beyond memorization.
> >
> > 2. The new information is simply contamination-free domain expertise measurement. Consider Llama-3.1 fine-tuned (FT) vs original Llama-3.1 against DeepSeek-v3: FT cannot translate QA accuracy gains into debate expertise. Since models unavoidably have varying contamination levels (difficult to reverse-engineer), measuring genuine expertise becomes our novel contribution.
> >
> > Our case studies reveal clear patterns: Q7700: FT defends $1.44 with excessive repetition; as Con, loops calculating the same answer, unable to propose alternatives. Q8005: FT's mathematical reasoning deteriorates—shallow defense, Con merely agrees "correct answer is 4." Q8013: FT shows statistical concept degradation—generic claims when defending, one-line capitulation as Con.
> >
> > The pattern is striking: contamination improves memorized defense but not the genuine expertise that debate uniquely measures beyond surface accuracy.
> >
> > ---
> >
> > Thank you again for your engagement with our work—your questions have helped us clarify key aspects of our approach!

---

### Author Response · Authors · 2025-05-31
**Additional Experimental Results**

We thank all reviewers for their thoughtful and constructive feedback! Inspired by your comments, we conducted additional large-scale experiments to further validate our approach.

### **New Experiment 1**: Full GPQA Dataset Evaluation

To address concerns about sample size and demonstrate generalization beyond MMLU-Pro, we allocated resources and evaluated our method on the complete GPQA main test set (Rein et al., 2023) (448 questions) using 6 open-source models:

- Phi-4 Multimodal (5.57B), Qwen3-32B, Llama-4-Scout, Llama-3.1-8B, Mixtral-8×7B, Mistral-7B

**Key Results:**

- **No transitivity violations** in overall win rates (combined Pro/Con roles)
- Clear performance hierarchy maintained across all model pairs
- Results confirm our method's robustness **across different QA datasets**

**Win Rate Matrix (Overall):**

| Model | Qwen3-32B | Llama-4-Scout | Llama-3.1-8B | Mixtral-8×7B | Mistral-7B | Phi-4 MM |
| --- | --- | --- | --- | --- | --- | --- |
| Qwen3-32B | 0.50 | 0.58 | 0.77 | 0.75 | 0.76 | 0.79 |
| Llama-4-Scout | 0.42 | 0.50 | 0.78 | 0.83 | 0.80 | 0.87 |
| Llama-3.1-8B | 0.23 | 0.22 | 0.50 | 0.51 | 0.61 | 0.83 |
| Mixtral-8×7B | 0.25 | 0.17 | 0.49 | 0.50 | 0.56 | 0.82 |
| Mistral-7B | 0.24 | 0.20 | 0.39 | 0.44 | 0.50 | 0.71 |
| Phi-4 MM | 0.21 | 0.13 | 0.17 | 0.18 | 0.29 | 0.50 |

### **New Experiment 2**: Contaminated Judge Analysis (Expansion of Section 5.5, Table 3)

We tested whether contaminated judges could compromise evaluation fairness by adding two new judge models to our original analysis: Llama-3.1-8B and Llama-3.1-8B-FT (fine-tuned on test set).

**Win Counts by Judge Model:**

| Debater | Llama-3.1-8B (new) | Llama-3.1-8B-FT (new) | Mistral Large | GPT-4o | ... |
| --- | --- | --- | --- | --- | --- |
| Mistral Large | 167 | 169 | 209 | 215 | (same as Table 3) |
| GPT-4o Mini | 162 | 167 | 176 | 191 | ... |
| Mixtral-8×7B | 144 | 147 | 122 | 126 | ... |
| Mistral-7B | 127 | 117 | 93 | 68 | ... |

**Win Rate Matrices for New Judges:**

*Llama-3.1-8B as Judge:*

| Debater | Mistral Large | GPT-4o Mini | Mixtral-8×7B | Mistral-7B |
| --- | --- | --- | --- | --- |
| Mistral Large | 0.50 | 0.52 | 0.58 | 0.57 |
| GPT-4o Mini | 0.48 | 0.50 | 0.56 | 0.58 |
| Mixtral-8×7B | 0.42 | 0.44 | 0.50 | 0.58 |
| Mistral-7B | 0.43 | 0.42 | 0.42 | 0.50 |

*Llama-3.1-8B-FT as Judge:*

| Debater | Mistral Large | GPT-4o Mini | Mixtral-8×7B | Mistral-7B |
| --- | --- | --- | --- | --- |
| Mistral Large | 0.50 | 0.51 | 0.54 | 0.64 |
| GPT-4o Mini | 0.49 | 0.50 | 0.56 | 0.62 |
| Mixtral-8×7B | 0.46 | 0.44 | 0.50 | 0.57 |
| Mistral-7B | 0.36 | 0.38 | 0.43 | 0.50 |

**Key Findings:**

- Both new judges produced consistent rankings: Mistral Large > GPT-4o Mini > Mixtral-8×7B > Mistral-7B
- **No transitivity violations** observed with either judge
- Results suggest role-switching protocol ensures fair evaluation, including resistance to potential positive-side bias from contaminated judges

**References:**

- [GPQA: A Graduate-Level Google-Proof Q&A Benchmark](https://arxiv.org/abs/2311.12022) (Rein et al., 2023)

---

### Decision · Program_Chairs · 2025-07-08

**Decision:**

Accept

**Comment:**

This paper proposes a debate-driven evaluation paradigm that can be applied to standard QA benchmarks. The results show that while fine-tuned models can show better QA accuracy, it is likely due to data contamination, because the same models perform worse in the structured debate setting. The proposed evaluation produces a robust ranking of models and is a promising way to address benchmark saturation.

**Strengths**

Reviewers point out that:

* The paper is well-written (wZxD) and clearly and compellingly motivated (wZxD, SpHc)
* The proposed evaluation method is effective and robust (5Sib)
* The claim is substantially supported by the experimental results (all reviewers)
* The finding about fine-tuning $\Rightarrow$ better standard benchmark performance but worse debate performance is especially interesting and compelling (wZxD, SpHc, 5Sib)

**Weaknesses**

Reviewers' biggest concern is the viability of the proposed method: debate requires much more LLM use than standard evaluation, and thus is substantially more costly and time-consuming (SpHc, 5Sib, fs1s). The rebuttal argues that in the long run creating hard-enough benchmarks will grow more expensive, while compute gets cheaper over time, alleviating this concern. I am mostly convinced by the rebuttal (and the reviewers raising their scores suggests that they are too), although I agree that it is too early to judge the practical impact of the approach.

Other criticisms had to do with the validity and reliability of the method:

* It was not demonstrated that success in debates aligns with human judgments about the performance/soundness of the reasoning (wZxD). I understand the rebuttal's argument about human expert evaluation being costly to conduct, but it would have further strengthened the paper.
* It was not originally clear if the findings generalize to other datasets (fs1s), but the additional experiments in the rebuttal addressed this.

Overall, echoing Reviewer SpHc: this work is a promising first step in an important research direction. Even if we can't guarantee its viability and future impact, I consider it a sufficient contribution and recommend acceptance -- given that the new results/discussion from the rebuttal are added to the paper.